# SAM-GAN: Supervised Learning-Based Aerial Image-to-Map Translation via Generative Adversarial Networks

Jian Xu ⬤, Xiaowen Zhou, Chaolin Han, Bing Dong and Hongwei Li *

School of Geoscience and Technology, Zhengzhou University, Zhengzhou 450001, China;
xj0102@gs.zzu.edu.cn (J.X.)
* Correspondence: lhw29691518@zzu.edu.cn; Tel.: +86-136-7371-2015

**Abstract:** Accurate translation of aerial imagery to maps is a direction of great value and challenge in mapping, a method of generating maps that does not require using vector data as traditional mapping methods do. The tremendous progress made in recent years in image translation based on generative adversarial networks has led to rapid progress in aerial image-to-map translation. Still, the generated results could be better regarding quality, accuracy, and visual impact. This paper proposes a supervised model (SAM-GAN) based on generative adversarial networks (GAN) to improve the performance of aerial image-to-map translation. In the model, we introduce a new generator and multi-scale discriminator. The generator is a conditional GAN model that extracts the content and style space from aerial images and maps and learns to generalize the patterns of aerial image-to-map style transformation. We introduce image style loss and topological consistency loss to improve the model's pixel-level accuracy and topological performance. Furthermore, using the Maps dataset, a comprehensive qualitative and quantitative comparison is made between the SAM-GAN model and previous methods used for aerial image-to-map translation in combination with excellent evaluation metrics. Experiments showed that SAM-GAN outperformed existing methods in both quantitative and qualitative results.

**Keywords:** generative adversarial networks; map generation; image-to-map translation; quality assessment; aerial imagery

## 1. Introduction

In modern life, maps are relevant to us. From general geo-information location search to in-car navigation, maps have become indispensable to human society. Traditionally, in the geographic information industry, maps are first measured in the field and then vectorized using professional mapping tools, which involves much time and repetitive human resources. Getting maps corresponding to the current situation in real-time is still challenging. Although various map vendors (Google Maps (Google Inc. in Mountain View, CA, USA), Baidu Maps (Baidu Inc. in Beijing, China)) already provide massive amounts of map information, there is a considerable iteration time gap as map vendors rely on manual collection to update map data. Therefore, this does not meet the needs of individual users. However, with the development of deep learning in the direction of image migration, a completely different means of mapping has emerged. In a review of the development of cartography in China, Liao Ke [1] suggests that future map mapping will be more intelligent and personalized. To realize the above needs, many scholars have used aerial or remote sensing imagery combined with image migration techniques to generate maps in an automated manner. This work makes it possible to generate maps without needing vector data. The success of such models will not only introduce new cartographic ideas to the field of map generation but also promote further development of the field of map mapping towards personalization and intelligence.

Recent image-to-image translation algorithms for map generation based on aerial imagery have achieved great success. However, there is still massive room for image accuracy and quality improvement.

In recent years, adversarial generative models have flourished and have had good results in various fields. Previous map generation models for aerial imagery did not focus on learning the target domain style from the model input level. The popular approach in the past was to guide the model to generate the target domain style results through pixel-level loss. Still, in this paper, these were generated by encoding the content and style of the aerial image and the map separately, i.e., the generator of the generative model includes not only the content encoder but also the style encoder. This allows the generated map to learn the map style better and faster while retaining the aerial image content. In addition, current generative models proposed in this field do not handle the topological relationships in the generated maps well. Our model adds topological consistency loss to bring about a significant visual improvement in the model results. In this paper, a deep learning model based on adversarial mechanisms aims to find a framework for aerial image-to-map transformation based on generative adversarial networks. We propose a novel supervised learning-based generative adversarial network for aerial image-to-map conversion. The model improves the applicability of map generation models by solving the problems of inadequate learning of map styles and topological integrity of map generation models. The model achieves generalization from aerial images of unknown regions to target map classes by learning the potential encoding space of content and styles extracted from aerial images and maps. Not only does our model include topological consistency constraints and style encoders, but we also find that adding an attention mechanism to the adversarial generation model can motivate the encoders to acquire more critical information when learning the potential encoding space corresponding to the images.

In this paper, we learn the content and style of images in different domain spaces and introduce attention mechanisms into the GAN architecture, proposing a novel supervised learning-based generative adversarial network for aerial image-to-map translation (in the paper, we will use SAM-GAN as a shorthand for the model) that aims to improve the quality and accuracy of the aerial image-to-map translation. Unlike traditional models, the generator of SAM-GAN has three modules: style encoder, content encoder, and decoder, while using multi-scale discriminators to integrate image information of different sizes to increase the guidance capability of the generator. Our network uses paired datasets, and this model performs better in terms of accuracy and visual perception by comparing it with the previous SOTA model.

## 2. Related Work

We divide the current work into three aspects: Firstly, the current state of research on image-to-image translation algorithms based on generative adversarial networks; secondly, the role of attention mechanisms for image conversion models; finally, the current state of development of aerial image-to-map translation work.

Image-to-image translation: In recent years, GAN [2] and image translation techniques have been used to generate images for a variety of purposes, in directions such as data enhancement [3,4], road extraction [5], and super-resolution images [6,7].

As a model with "infinite" generative power, a direct application of GAN is modeling, generating data samples consistent with real data distribution. Thus, GAN-based image-to-image translation has continued to achieve compelling results [8]. The pix2pix proposed by Phillip Isola [9] learns the mapping relationship between two domains using paired images and achieves better results. One of the most classical algorithms in unpaired inter-image translation, the CycleGAN model [10], learns the mapping relationship between the original and target domains well using cyclic consistency loss. The contemporaneous DualGAN [11] and DiscoGAN also illuminate the non-pairing domain. DualGAN uses the idea of coupling to extend the native GAN into two mutually coupled GANs, which provide substantial performance gains compared to a single GAN. DiscoGAN, proposed by Taeksoo

Kim et al. [12], learns the relationship between different domains and then uses the learned domain relationships to successfully transfer styles from one domain to another while retaining the key attributes of the original domain. However, the generated results of these models applied to aerial imagery and map transformation have not been able to achieve better visual results. The subsequent StarGANv2 proposed by Yunjey Choi et al. [13] aims to generate a single framework for image diversity and multi-domain scalability.

Recently, the UNIT [14] model based on the assumption that two images with the same semantic content but different style domain spaces have the same latent space vector has achieved good results in unsupervised image transformation. Later, the MUNIT model [15] argues that the shared space can encode both the content space and the space in which they should differ, i.e., the style space, thus proposing a multimodal unsupervised image-to-image migration framework. It generates polymorphic images by encoding different combinations of styles. The FUNIT [16] model, based on UNIT, achieves better results and improves the generalization ability of the model when trained on small- sample datasets.

Attention mechanisms in deep learning: The attention model comes from the study of human vision, where humans selectively focus on parts of the comprehensive information, which is the attention mechanism. Similarly, attention can be added to deep learning methods to motivate the model to assign weights to different regions to learn critical information and thus improve the model's performance. The attention mechanism has demonstrated its performance in various tasks, such as machine translation [17,18], image generation [19,20], and image classification [21–23].

The plug-and-play performance of the attention mechanism has made it widely used in deep learning applications and has minimal impact on model overhead. Volodymyr Mnih [24] combined recurrent neural networks with visual attention to improve image classification accuracy, and attention mechanisms are gaining popularity. Ashish Vaswani [25] used an attention mechanism to achieve excellent results in machine translation. The SeNet [26] model proposed by the autonomous driving company Momenta proposes the general-purpose channel attention module SeBlock, which makes better use of inter-channel correlation by both compression and excitation, and achieves outstanding performance in image recognition. CBAM [27] proposes a lightweight, general-purpose module combining spatial attention and channel attention, which can be used with convolutional neural networks for end-to-end training, improving model classification and detection performance with minimal overhead. ECA-Net [28] implements a local cross-channel interaction strategy without dimensionality reduction and adaptive selection of one-dimensional convolutional kernel size on top of SeNet, thus achieving improved performance.

Recent studies have found that incorporating attention mechanisms into generative adversarial network models can also achieve good performance. Self-AttentionGAN [19] uses spectral normalization and self-attentive mechanisms to learn the interdependencies between global features better, combining self-attentive mechanisms and convolutional neural networks to model long-range multi-level image regions, providing a method to combine global information. It is a better solution to the problem that GANs do not quickly learn specific structural and geometric features. Attention-GAN [29] decomposes the generator into two separate networks. The model will acquire the attention graph and then fuse the attention graph with the original image to obtain the final generated result. Hajar Emami's proposed SPA-GAN model [20] improves the model effect by using a discriminator to get the attention graph back to the generator, thus achieving better results on unpaired datasets.

Style translation of aerial images-to-maps: Inspired by generative adversarial networks, deep learning adversarial mechanisms to learn mapping patterns between aerial images and maps started to attract attention in map mapping. The data used in this paper is the Maps dataset (http://efrosgans.eecs.berkeley.edu/pix2pix/datasets/maps.tar.gz (accessed on 10 February 2023)) [9]. In that related work, when the models involved use that dataset, we provide a simple and clear comparative analysis of their performance on that dataset. Jun Gu et al. [30] better extracted feature edge information using a generator

architecture with cyclic consistency loss and pix2pixHD [31]. However, combined with the model's performance on the Maps dataset, we argue that the model only combines the generator of the Pix2pixHD model and the discriminator along the lines of the Pix2pix model [9], and does not improve the quality of the generated maps any better. GeoGAN [32] proposed a RealNVP bijection model based on CGAN [33] that can better learn the style space of the target domain, although the RealNVP flow model idea used in this model presents a new perspective on the research problem of this paper. However, in combination with the model's results on the dataset Maps, the feature boundaries of the generated maps are very blurred. Furthermore, the generated maps are poorly rendered in color due to insufficient learning of the map domain. Users are unable to obtain useful information from the generated maps. The SG-GAN [34] model later rendered the crowdsourced GPS data into images using layers and integrated them into GAN while combining semantic rules to estimate the high-level information of the images, significantly improving the road in image migration accuracy. However, it requires a large amount of GPS data, which may affect the model's usefulness. On the other hand, the MapGAN model [35] uses a rendering matrix to focus on geographic entity attributes and aesthetic color rendering to give a better visual effect to the generated map. Although MapGAN has improved the performance of the generated maps in terms of color rendering, it can lead to overfitting problems with the model. As shown by its performance on the Maps dataset, it does not solve the problem of generating maps with clear boundaries between feature boundaries nor does it guarantee the accuracy of the color rendering locations.

## 3. Materials and Methods

In this section, we first present the proposed SAM-GAN's general architecture and then detail the model's components. Finally, we explain the role of each loss function used in the learning of the model.

### 3.1. The Core of SAM-GAN

We propose SAM-GAN, a supervised learning-based architecture for transforming aerial images and maps. Figure 1 shows the core idea of SAM-GAN, which is inherited from the UNIT [14] model. It indicates that the potential space of an image can be decomposed into a content space and a style space where the content space represents the content of the source domain and the style space represents the style of the target domain. By decomposing and selectively combining the content and style spaces from different domains, the model is motivated to consider both the content space of the source domain and the style space of the target domain. In this paper, the source domain refers to aerial images, and the target domain refers to maps.

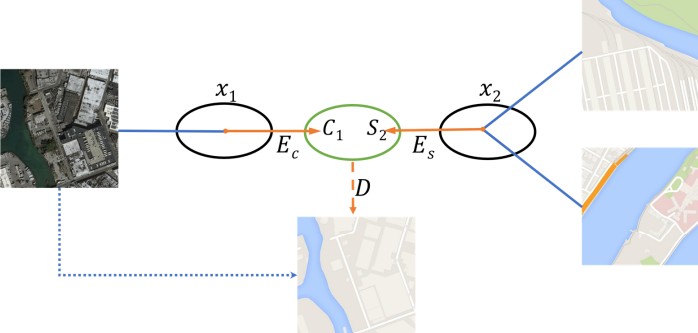

$x$ represents a set of original domains.
$E_c$ represents the content encoder, $E_s$ represents the style encoder.
$D$ represents the decode.
$S$ represents the style space of the domain.
$C$ represents the content space of the domain.
The arrow points to the map(Corresponding to aerial) representing the model generation.

**Figure 1.** The core of the SAM-GAN model. $x_1$ represents the source domain (aerial image), and $x_2$ represents the target domain (map). $C_1$ represents the content space of the aerial image, and $S_2$ represents the style space of the map. Different types of encoders can separate the content space or style space from the domain, while decoders must combine the content space and style space.

### 3.2. The Overall Architecture of SAM-GAN

Our model aims to train generators to learn mapping relationships between source and target domains. The SAM-GAN model is divided into two main parts: a generator and a discriminator, where the generator consists of a content encoder, a style encoder, and a decoder. Similar to other GAN problems, SAM-GAN needs to have excellent capabilities in the training and testing process to transform unlearned source domain images well into images. In a data form, we use x to represent aerial images and $y$ to represent map images, where y denotes a collection of K maps in the target domain, i.e., $y = \{y_1, y_2, \ldots y_K\}$.

Figure 2 shows the detailed architecture of the model. The input to the model is divided into two parts: the first part is an aerial image, and the second is *K* maps, where the aerial image is used in the content encoder to extract the content space. In contrast, the maps are fed into the style encoder to learn the style space of the target domain, resulting in a model architecture suitable for two-domain transformation. One of the things to note is that the order of the images input to the model during training is random. The discriminator guides the generator during training by learning the distribution of images in the source and target domains, thus allowing the generator to generate a more realistic map. In model testing, the model is fed with aerial images that have never been seen in training. The model must generate the corresponding maps when we provide the completed model with aerial images from the source domain.

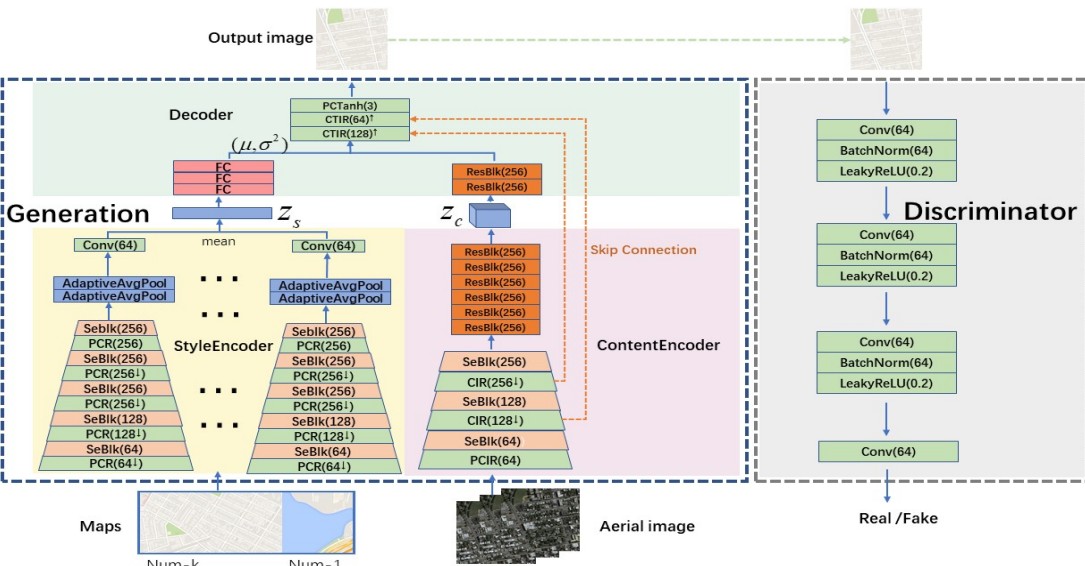

**Figure 2.** Detailed architecture of the SAM-GAN model, where PCR represents the combination of Padding, Convolution, and ReLU activation functions; SeBlk represents the attention module, PCIR represents the combination of Padding, Convolution, Instance Normalization, and ReLU activation functions; FC represents the fully connected layer; CTIR represents the Transpose Convolution, Instance Normalization, and ReLU combination; $Z_S$ denotes the style potential coding vector, and $Z_C$ denotes the content potential coding feature map.

### 3.2.1. Generators

The generator of SAM-GAN, an aerial image-to-map translation model, consists of three parts: a content encoder, a style encoder, and a decoder.

Unlike previous aerial image-to-map translation methods, SAM-GAN introduces a style encoder to learn the style space of the target domain. The input of the style encoder contains K map images from the target domain; each map is passed through the style encoder to obtain a one-dimensional vector, and the final output of the style encoder is obtained by averaging the K potential style encoding vectors. The result represents the potential coding vectors in the style space of the map domain, which in turn guides the decoder to work better.

In addition to the fully connected layer, the content encoder and the decoder are based on an improved Pix2pix [9] model architecture. The content encoder consists of four parts: the input convolution block, the attention block, the downsampling block, and the residual block. The decoder also has four components: the residual block, the upsampling block, the output convolution block, and the multilayer perceptron. As shown in Figure 2, PCIR denotes the input convolutional layer, CIR denotes the downsampling layer, SeBlk [26] denotes the attentional layer, ResBlk denotes the residual block, CTIR denotes the upsampling layer, and PCTanh denotes the output convolutional layer. Three consecutive FC blocks denote a multilayer perceptron. The main improvements include three aspects:

1. Adding the channel attention mechanism SeBlock (abbreviated as SeBlk) at the beginning of the encoder, which can obtain the importance level of each channel in the feature map, then use this importance level to assign the corresponding weight value to each feature, thus allowing the model to focus on specific feature channels and enhance the channel importance of certain feature maps, which ultimately allows the model to better filter out important information in a large amount of primary.

2. We use nine residual blocks, of which seven are used in the content encoder and two in the decoder using adaptive instance normalization, which we call adaptive residual blocks, where the initial weights of the adaptive residual blocks come from

the multilayer perceptron. Adaptive residual blocks enable the decoder to combine the content space obtained from the content encoder and the style space obtained from the content encoder and the style encoder to generate high-quality translation results.

3. We establish connections at the channel level between the content encoder and the decoder via jump connections which allows the model to better combine low-level features from earlier stages with high-level semantic features from later stages to achieve cross-layer information propagation and serves to prevent the model from gradient disappearance during the training process.

4. In the decoder part of SAM-GAN, the adaptive instance normalization (AdaIN) [36] method that we use is a fundamental reason for the decoder to make efficient use of the style space learned by the style encoder. AdaIN works in detail as follows: each channel of the latent content encoding feature map $Z_C$ obtained from the content encoder is first normalized, during which a multi-layer perceptron consisting of three fully connected layers of the decoder computes the style latent encoding vector $Z_S$ from the style encoder to obtain the mean and variance vectors ($\mu$,$\sigma_2$); then, $\mu$ is applied to the adaptive residual block of the decoder as a deviation and $\sigma_2$ as a scaling factor to jointly parameterize the affine transformation.

### 3.2.2. Discriminator

The discriminator idea of the SAM-GAN model comes from the PatchGAN [9] architecture, which uses a receptive field of size $70 \times 70$ to judge the resulting feature map. The specific discriminator architecture is shown in the right-hand area of Figure 2. During the model's training, the discriminator not only learns aerial and map images but also feeds the results back to the generator to guide its generation process.

### *3.3. Loss Function*

For SAM-GAN to efficiently understand the coding space learned by content and style encoders, our model combines the following loss terms: adversarial loss, content loss, and style loss. They ultimately constitute the total loss function of the model.

### 3.3.1. Adversarial Loss

The model SAM-GAN aims to convert aerial images into maps, where the generator's purpose is to generate realistic maps through the learned style space, while the purpose of the discriminator is to distinguish between the actual and false maps of the input. Thus, the generator minimizes the adversarial loss, while the discriminator maximizes the objective function. The two form a game relationship in which the model moves towards equilibrium in a constant adversarial process. The equation can therefore be expressed as

$$\min_{G} \max_{D} L_{GAN}(G,D) = E_{y \sim P_{(y)}}[\log D(y)] + E_{x \sim P_{(x)}, y \sim P(y)}[\log(1 - D(G(x,y)))] \tag{1}$$

### 3.3.2. Content Loss

The content loss of the SAM-GAN model consists of three aspects: VGG loss, L1 pixel-level loss, and topological consistency loss.

The VGG loss [37] compares the gap between the generated image and its corresponding real map. The loss is achieved using a pre-trained VGG-19 model [38] to extract features from the style-transformed image and its corresponding real map, obtaining multiple feature maps corresponding to both. Then we use the L1 loss function and set different weights to calculate the gap between the corresponding feature maps, which is expressed as

$$L_{VGG} = E_{x \sim P_{(x)}, y \sim P(y)} \left( \sum_{i,j} \|F_{ij}^l - P_{ij}^l\|_1 \right) \tag{2}$$

where $N^l$ is used to denote the number of channels contained in the lth feature map, $M^l$ denotes the $N^l$ feature map contained in the lth feature map, and $M^l$ is the height of

the feature map multiplied by its width so that the lth feature map can be expressed as $F^l \in \mathcal{R}^{N^l \times M^l}$, where $F^l_{ij}$ denotes the feature value of the $j$th position of the $i$th channel in the $l$th feature layer.

The $L_1$ pixel-level loss [39] is a direct calculation of the $L_1$ loss between the style translation result and the real map, as follows:

$$\mathrm{L}_{pixel} = E_{x \sim P_{(x)}, y \sim P(y)}(\|G(x) - y\|_1) \tag{3}$$

In geography, topological relations reflect the logical relationships between spatial entities and describe the connections between lines and planes in space. So, a map that can express better topological information is one of the prerequisites for its practical application. Topological consistency loss is added to the optimization objective in SAM-GAN to have better structural characteristics of the elements in the generated map. The topological consistency loss refers to the content loss function of the SMAPGAN model [40]. In maps, features are mainly composed of points and lines, and the topological relationships between entities can be represented using their edges, whereas in digital image processing, gradients can be used for edge detection. Thus, we can use image gradients to represent the edges of image features and thus express the topological relationships between map elements.

Let the pixel value of a pixel point $(i, j)$ of the image be $H(i, j)$. The horizontal gradient of this pixel point is

$$v_i(i, j) = H(i + 1, j) - H(i - 1, j) \tag{4}$$

The vertical gradients are

$$v_j(i, j) = H(i, j + 1) - H(i, j - 1) \tag{5}$$

Therefore, the gradient of the image at the pixel point is known to be

$$\mathrm{L}_{top} = E_{x \sim P_{(x)}, y \sim P(y)}(\|v(G(x)) - v(y)\|_1) \tag{6}$$

The loss of the elements comprising the above three areas is therefore

$$\mathrm{L}_{content} = \alpha_1 \mathrm{L}_{vgg} + \alpha_2 \mathrm{L}_{pixel} + \alpha_3 \mathrm{L}_{top} \tag{7}$$

where $\alpha_1$, $\alpha_2$ and $\alpha_3$ represent the weights of the corresponding loss terms.

### 3.3.3. Style Loss

Gatys [41] proposed that the representation of content and style in a convolutional neural network is separable and that the image has a good target domain style after translation by using style loss to measure the style before and after the image style translation. We input the style-transformed image with its corresponding real map into the VGG-19 network, extract each feature map layer, and calculate the corresponding Gram-style matrix [42]. The Gram-style matrix can describe the correlation between features and also possesses the feature of positional insensitivity. Hence, we use the Gram-style matrix as a quantified description of image style. Since one feature map corresponds to one Gram-style matrix, for multiple feature maps, we use the set $\{GM^1, GM^2, \ldots, GM^l\}$ to denote the set of Gram-style matrices corresponding to the feature map $F$. We use the L1 loss function, which is represented by the following equation:

$$GM^l_{ij} = \sum_k F^l_{ik} F^l_{jk} \tag{8}$$

$$L_{style} = E_{x \sim P_{(x)}, y \sim P(y)}\left(\left\|\frac{1}{N^l M^l} GM^l(G(x)) - GM^l(y)\right\|_1\right) \tag{9}$$

## 4. Results

In this section, we first describe the sources of the dataset. The model is then compared quantitatively using four objective evaluation metrics. Then, the qualitative comparison phase illustrates the performance of the SAM-GAN-generated results in terms of visual perception using three subjective evaluation metrics. Finally, we perform ablation experiments to analyze the functionality of the model components.

### 4.1. Dataset and Experimental Setups

The Maps dataset [9] was created from satellite and electronic map tiles from Google Maps, with an image size of $600 \times 600$ pixels. It consists of 2194 data pairs, each consisting of an aerial image and its corresponding map tile image extracted from Google Maps. The sample area included in the dataset is New York City and its surrounding area, and the aerial imagery has a spatial resolution of 2.15 m/pixel in the visible band.

We used the PyTorch (version 1.9.0) framework to implement our model. In terms of hardware, an NVIDIA GeForce RTX 3090 graphics card with 24 GB of video memory was used for training. We choose the Adam optimizer in the model training details and set $\beta_1 = 0.9$, $\beta_2 = 0.999$. We also set the initial learning rate to 0.0002 and set it to decrease by half every 50 cycles. The training batch size of the model is 4, the size of the model input and output images are both $256 \times 256$, and the weights in the content loss function are $\alpha_1 = 10$, $\alpha_2 = 10$, and $\alpha_3 = 1$.

To fairly compare the performance of each method, all models in the experiment use the same image pre-processing process in the training process.

### 4.2. Baselines

We chose four SOTA methods to compare with the model proposed in this paper: AttentionGAN [29], CycleGAN [14], Pix2pix [9], and UNIT [14]. Among them, pix2pix requires paired datasets for training, while AttentionGAN, CycleGAN, and UNIT are unsupervised training models and do not require paired datasets.

Pix2pix, an image transformation model based on the CGAN model, was used as one of the baseline models for this experiment due to the generality of its architecture and its better performance in supervised learning of image-to-image transformations.

CycleGAN, as one of the classical unsupervised learning architectures for image transformation models, proposes a cyclic consistency loss that can still perform image transformation between two domains in the absence of paired data, which first attempts to map images from the source domain to the target domain and then back to the original domain, reducing the difference between the reconstructed image and the original domain while motivating the model to learn better the mapping relationship between the source and target domains. The relationship between the source and target domains is reduced.

AttentionGAN minimizes the difference between the source and target domains in the data generation distribution by adding an attention mechanism to CycleGAN.

The UNIT model is a novel unsupervised image-to-image translation model that learns the spatial distribution of the original domain image by separating its content encoding from the style encoding of the target domain image, which in turn fuses to obtain the converted target domain image eventually.

### 4.3. Evaluation Metrics

In our experiments, we combined objective and subjective evaluations to assess the maps generated by the model. We have paired data so that the generated images have their counterparts in real maps. We considered the images' quality, diversity, and accuracy in choosing the evaluation metrics. For the quantitative evaluation, we chose four evaluation metrics: inception score [43], Fréchet inception distance [44], structural similarity [45], and pixel accuracy [46]. For the qualitative evaluation, we use three metrics: content retention, style similarity, and map availability [47].

### 4.3.1. Inception Score (IS)

IS is a class of metrics used to assess the quality of the generated images. It evaluates the quality of images in two main aspects: sharpness and diversity. In the comparison experiments, we calculated the IS scores of the images generated by each model on the test set and then compared them. The higher the IS score, the better the quality of the images generated by the model and the better the model's generalization ability.

### 4.3.2. Fréchet Inception Distance (FID)

As the IS metric only evaluates image quality, we introduced the FID to reflect the distance between the real and generated images. The lower the FID score, the better, which indicates that the model generates image results that are more similar to the real image.

### 4.3.3. Structural Similarity (SSIM)

The purpose of SSIM is to measure the similarity index between two images. It considers three main characteristics of an image: brightness, contrast, and structure. As a perception-based evaluation metric, SSIM is visually more aligned with human intuition. The formula is as follows:

$$\text{SSIM}(x, y) = \frac{\left(2u_x \mu_y + c_1\right)\left(2\sigma_{xy} + c_2\right)}{\left(u_x^2 + u_y^2 + c_1\right)\left(\sigma_x^2 + \sigma_y^2 + c_2\right)} \tag{10}$$

where $x$ is the image obtained by the generator and $y$ is the real map image corresponding to $x$. $\mu_x$ is the mean of $x$, $\mu_y$ is the mean of $y$, $\sigma_x^2$ is the variance of $x$, $\sigma_y^2$ is the variance of $y$, and $\sigma_{xy}$ is the covariance of $x$ and $y$. $c_1 = (k_1 L)^2$ and $c_2 = (k_2 L)^2$ are constants used to maintain stability ($k_1 = 0.01$, $k_2 = 0.03$, and $L$ is the dynamic range of pixel values). The range of SSIM is $[-1, 1]$, and the value of SSIM is equal to 1 when the two images are identical.

### 4.3.4. Pixel Accuracy

The abbreviation for pixel accuracy is PixelACC (%). It is implemented as follows: first, the RGB values $(r_i, g_i, b_i)$ of the original map and the RGB values $(r'_i, g'_i, b'_i)$ of the generated map are obtained. Then the difference between each channel is calculated, and the maximum difference is compared with the threshold $\delta$. If the result is less than $\delta$, the generated image is considered similar to the real map and generated accurately.

$$\left(\left|r_i - r'_i\right| + \left|g_i - g'_i\right| + \left|b_i - b'_i\right|\right) < \delta \tag{11}$$

In this paper, we take $\delta$ to be 1 and 5.

### *4.4. Comparisons with Baselines*

We used objective and subjective evaluation metrics to compare the model proposed in this paper with the four baseline models mentioned above, thus demonstrating the effectiveness of SAM-GAN in the aerial image and map translation.

### 4.4.1. Objective Evaluation

This section compares the SAM-GAN model with each baseline model to illustrate the SAM-GAN model's powerful capabilities. In the comparison experiments, we set the generator's style encoder parameter $K$ equal to 1 and tested the translation results of SAM-GAN and the baseline model on the same test set using four different image quality evaluation metrics to illustrate the generalization capabilities of each model. Table 1 shows the quantitative comparison between the baseline model and SAM-GAN. Table 2 gives the training time required for each model.

**Table 1.** Quantitative results of the baseline model and SAM-GAN ($K = 1$) on the performance of aerial image-to-map translation for different evaluation metrics, where $\delta$ is taken as 5. (Bold: performance first; underlined: performance second).

| Models | IS | FID | SSIM | PixelACC (%) |
|---|---|---|---|---|
| AttentionGAN | 2.9454 | <u>82.0007</u> | 0.1958 | <u>30.8973</u> |
| CycleGAN | 2.6103 | 211.1872 | 0.2993 | 24.2836 |
| Pix2pix | <u>3.0706</u> | 136.0318 | <u>0.3990</u> | 27.7821 |
| SAM-GAN($K = 1$) | **3.7557** | **65.7382** | **0.4343** | **33.3638** |
| UNIT | 3.0576 | 149.9581 | 0.3252 | 23.9213 |

**Table 2.** Training time required for the baseline and SAM-GAN models ($K = 1$). (Bold: performance first; underlined: performance second).

| | AttentionGAN | CycleGAN | Pix2pix | SAM-GAN | UNIT |
|---|---|---|---|---|---|
| Time (h) | 9.56 | 7.73 | <u>3.33</u> | **1.21** | 30.27 |

The evaluation metric scores show that the SAM-GAN framework significantly outperforms the baseline model in the four metrics of IS, FID, SSIM, and PixelACC(%) in the aerial image-to-map translation task. Regarding IS, by comparing the IS values of the models, we found that SAM-GAN has the largest IS value, which indicates that SAM-GAN generates the highest-quality images. In terms of FID, SAM-GAN still has the best performance, with a two-fold improvement relative to the Pix2pix model and a three-fold improvement compared to CycleGAN. In terms of SSIM, SAM-GAN doubles its performance compared to the baseline model AttentionGAN. Regarding pixel accuracy, SAM-GAN achieves an accuracy of approximately 33.364% with parameter $\delta$ equal to 5, which is over 9% better than CycleGAN and nearly 10% better than UNIT. FID, SSIM, and pixel accuracy results show that SAM-GAN generates the closest results to the real map. Table 1 illustrates that previous image-to-image translation methods do not make good use of a priori knowledge to learn target classes better when applied to aerial image-to-map translations because these models cannot learn the style space and topological feature details of the target domain well. SAM-GAN uses content and style encoders to combine the content of the original domain with the style of the target and the attention module to focus on the essential parts of the different domain spaces while preserving the topology of the image to achieve a higher-quality image transformation.

The training time of the models given in Table 2 shows that the training time of Pix2pix using supervised learning strategies is in the range of 3–4 h, while the three models using unsupervised learning strategies, CycleGAN, AttentionGAN, and UNIT, take longer, with the first two of them having cyclic consistency loss taking around 7–10 h to train, while UNIT took the longest time of approximately 30 h. In contrast, the SAM-GAN model in this paper required the shortest training time, at 1–2 h. So SAM-GAN has higher translation results in terms of image quality and better model convergence and learning ability in the aerial image and map translation applications.

### 4.4.2. Subjective Evaluation

The results of the objective evaluation metrics show that the accuracy scores of the baseline models and SAM-GAN appear to be low. However, from a visual perspective, slight differences in accuracy do not have a decisive impact on using the maps. Therefore, to determine the visual effectiveness of the models, 25 people from the GIS profession with knowledge of map mapping were invited to qualitatively evaluate the output of the baseline and SAM-GAN models in terms of content retention, style similarity, and map usability. The scores for the three metrics take a range of 1–10, with higher scores representing the better performance of the image on that metric.

Content retention: indicates whether the generated map matches the content information of the corresponding aerial image; a higher score means the complete content retention of the generated map.

Style similarity: indicates the degree of similarity between the generated map and the corresponding real map; a higher score means that the generated map is more similar to the corresponding real map.

Map availability: indicates whether the generated map can be used in reality; a higher score represents its ability. Higher scores represent better information on features and the ability to express map elements in the generated map.

Table 3 shows a qualitative comparison between the baseline model and the SAM-GAN model, and Figure 3 shows the details of the images generated by the model. The comparison of the images in Figure 3 shows that the SAM-GAN model performs better in terms of the quality and structure of the generated results compared to the Pix2pix supervised learning method, e.g., the degree of correct distribution of colors corresponding to different features and the detail of buildings. Compared to the other three unsupervised learning methods, SAM-GAN performs better in terms of topological integrity of features, clarity of boundary lines between different features, and color distribution.

**Table 3.** Qualitative comparison results between the baseline model and SAM-GAN on the performance of aerial image-to-map translation. (Bold: performance first; underlined: performance second).

| Models | Content Retention | Style Similarity | Map Availability |
| --- | --- | --- | --- |
| AttentionGAN | <u>6.65</u> | <u>6.68</u> | <u>6.45</u> |
| CycleGAN | 5.63 | 5.60 | 5.46 |
| Pix2pix | 6.24 | 6.45 | 6.32 |
| SAM-GAN($K = 1$) | **7.94** | **8.05** | **7.77** |
| UNIT | 4.83 | 4.89 | 4.73 |

The data in Table 3 show that SAM-GAN is the best compared to the baseline models regarding content retention, style similarity, and map availability, indicating that the model is the best regarding subjective perception. The results for map availability show that although the SAM-GAN model does not score well in some quantitative evaluation metrics, this does not negate its potential use in some practical scenarios that require personalization scenarios, e.g., rescue and disaster relief.

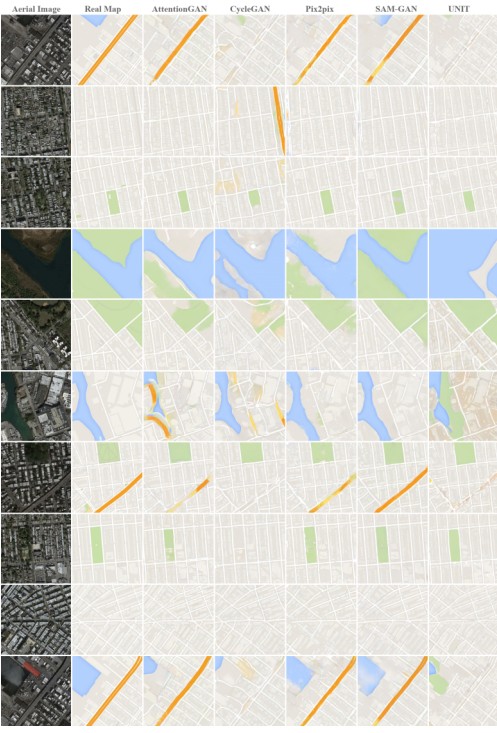

**Figure 3.** Results of SAM-GAN and each baseline model in terms of aerial image-to-map translation performance, from left to right, aerial images, real maps, results of AttentionGAN, results of CycleGAN, results of Pix2pix, results of SAM-GAN, and results of UNIT.

*4.5. Exploration of the Number of Style Encoder Input Maps K in Training*

In this section, the main focus is to explore whether there is a relationship between the ability of the model and the *K*-value. We plan to observe the model's performance by setting different *K* values with the help of four types of objective evaluation metrics. As shown in the table below, we input an aerial image to the content encoder in training, and *K* randomly selected real maps for the style encoder. We set the values of *K* to be 1, 3, 5, 8, and 10 and evaluate the scores of the model under different *K* values on the test set using four evaluation metrics.

Table 4 shows the model scores on each evaluation metric for different *K* values. Figures 4 and 5 show the trends of the evaluation metrics for different *K* values taken by the model, while Table 5 shows the time taken to train the model for different *K* values.

**Table 4.** Quantitative assessment scores of the SAM-GAN model for different *K* values where $\delta$ is taken as 5. (Bold: first in performance; underlined: second in performance).

| Models | IS | FID | SSIM | PixelACC(%) |
|---|---|---|---|---|
| SAM-GAN ($K = 1$) | **3.7557** | **65.7382** | <u>0.4343</u> | **32.5677** |
| SAM-GAN ($K = 3$) | 3.6737 | 71.8303 | 0.4340 | 32.5300 |
| SAM-GAN ($K = 5$) | <u>3.7021</u> | <u>71.1599</u> | **0.4353** | <u>32.5526</u> |
| SAM-GAN ($K = 7$) | 3.5394 | 71.3984 | 0.4311 | 32.4577 |
| SAM-GAN ($K = 10$) | 3.6217 | 71.4494 | 0.4299 | 32.5048 |

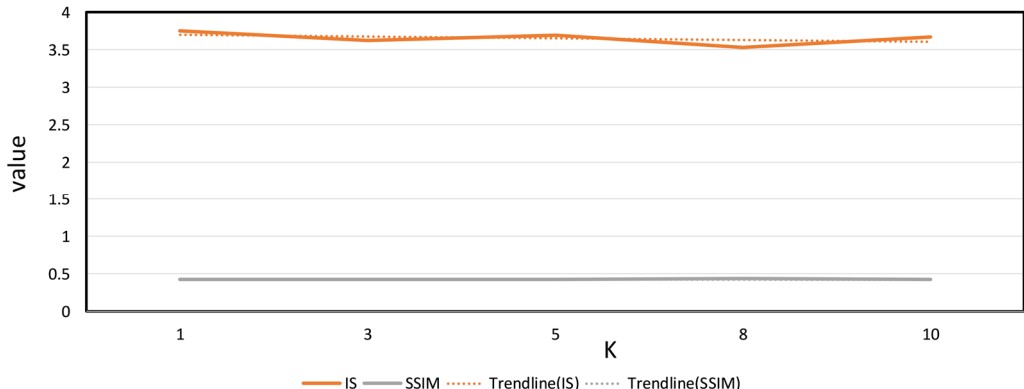

**Figure 4.** Trend plots of IS and SSIM scores for the SAM-GAN-*K* (*K* = 1, 3, 5, 8, 10) model on the test set.

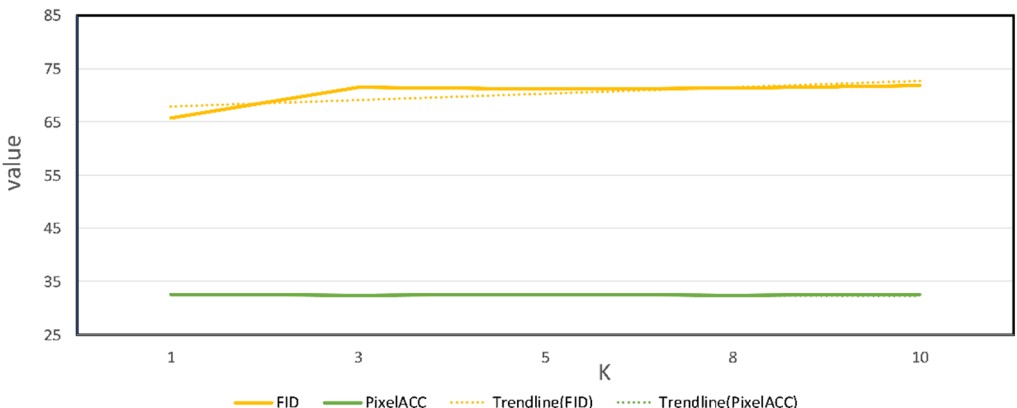

**Figure 5.** Trend of FID and PixelACC(%) scores for the SAM-GAN-*K* (*K* = 1, 3, 5, 8, 10) model on the test set.

**Table 5.** Time required to train the SAM-GAN model for different K values.

|  | *K* = 1 | *K* = 3 | *K* = 5 | *K* = 8 | *K* = 10 |
|---|---|---|---|---|---|
| Time(h) | 1.21 | 3.03 | 3.38 | 4.08 | 4.65 |

The scores in Table 4 show that the model achieves first place in IS, FID, and PixelACC (%) when *K* is taken as 1. Although the score of SSIM (*K* = 1) is smaller than that of SSIM (*K* = 5), the difference is only about 0.001. The trend of each evaluation metric with K shown in Figures 4 and 5 shows that the performance of SAM-GAN does not increase with *K*. There is a more obvious fluctuation in the number of domain classes. The training time of the model varies for different values of *K*. The *K* value is proportional to the training time of the model, and the model has the shortest training time and the best generalization ability on the test set when *K* is set to 1. Unlike traditional pairwise training models, the above findings demonstrate that the model in this paper can achieve fast style migration with small samples.

By combining SAM-GAN's results in quantitative evaluation and training time, we concluded that the number of maps input to the training process affects the performance of SAM-GAN, where the model performs best with a *K* range of 1–5. This also indicates no mutual influence between the content encoder learning image content and the style encoder learning image style in SAM-GAN training. The style encoder uses the learned target domain style encoding vector to guide the decoder in generating content images with the target domain style.

*4.6. Ablation Study*

To determine the plausibility of SAM-GAN for aerial image translation applications, we performed ablation tests where K was taken as 1. We validated the importance of each SAM-GAN component on the same test set. First, we kept all parts of the model. Second, we removed only the style encoder. Third, we removed only the VGG loss in content loss. Fourth, we removed only the topological consistency loss in content loss. Finally, we removed only the SeBlock module, the attention mechanism used. These ablation methods are described as follows:

- SAM-GAN-no-StyleEncoder: SAM-GAN has the style encoder removed from the generator.
- SAM-GAN-no-VGGLoss: SAM-GAN does not use VGG content loss.
- SAM-GAN-no-TOPLoss: SAM-GAN does not use topological consistency loss.
- SAM-GAN-no-SeBlock: the attention module is removed from the generator of SAM-GAN.

Table 6 shows the objective evaluation scores for each ablation experiment. Table 7 shows the corresponding training time for the model under each experiment. Figure 6 shows the generated results for each experiment on the dataset Maps.

**Table 6.** Quantitative assessment scores for SAM-GAN ablation experiments, where $\delta$ is taken as 1. (Bold: performance first; underlined: performance second).

| Metrics | IS | FID | SSIM | PixelACC (%) |
|---|---|---|---|---|
| SAM-GAN-no-StyleEncoder | 3.1914 | 97.2184 | 0.3927 | 0.2862 |
| SAM-GAN-no-VGGLoss | 3.4956 | 81.7214 | 0.4158 | 0.3237 |
| SAM-GAN-no-TOPLoss | 3.6013 | <u>70.1293</u> | 0.4288 | 0.3236 |
| SAM-GAN-no-SeBlock | <u>3.7019</u> | 71.4534 | <u>0.4298</u> | <u>0.3256</u> |
| SAM-GAN | **3.7557** | **65.7382** | **0.4343** | **0.3257** |

**Table 7.** Quantitative assessment scores for SAM-GAN ablation experiments, where $\delta$ is taken as 1. (Bold: performance first; underlined: performance second).

| | SAM-GAN-No-StyleEncoder | SAM-GAN-No-VGGLoss | SAM-GAN-No-TOPLoss | SAM-GAN-No-SeBlock | SAM-GAN |
|---|---|---|---|---|---|
| Time(h) | 3.52 | 3.68 | 3.31 | <u>2.37</u> | 1.21 |

Comparing the results in Table 6, the SAM-GAN-no-StyleEncoder with the missing style encoder has the worst results. At the same time, in Figure 6, it can be seen that this experiment corresponds to a substantial decrease in the quality of the generated images and the pixel-level accuracy of the images. This is followed by the performance of SAM-GAN-no-VGGLoss and SAM-GAN-no-TOPLoss, which differ from SAM-GAN in objective evaluation, mainly regarding the FID and SSIM metrics of image quality. In Figure 6, it can be seen that SAM-GAN-no-VGGLoss generates results with a larger gap in terms of the accuracy of the color and distribution of features in the images compared to SAM-GAN. The difference between SAM-GAN-no-TOPLoss and SAM-GAN is mainly in the topological integrity of the roads and the clarity of the boundary lines between features. For the channel attention mechanism used in the model, we find that SAM-GAN-no-SeBlock performance is the closest to SAM-GAN. By, combining the time required for training each ablation experiment provided in Table 7, we know that SAM-GAN-no-SeBlock is second only to SAM-GAN in terms of time needed; despite this, the time cost of the former is close to twice the time cost of the latter, which indicates that SeBlock acts as an accelerator for model learning.

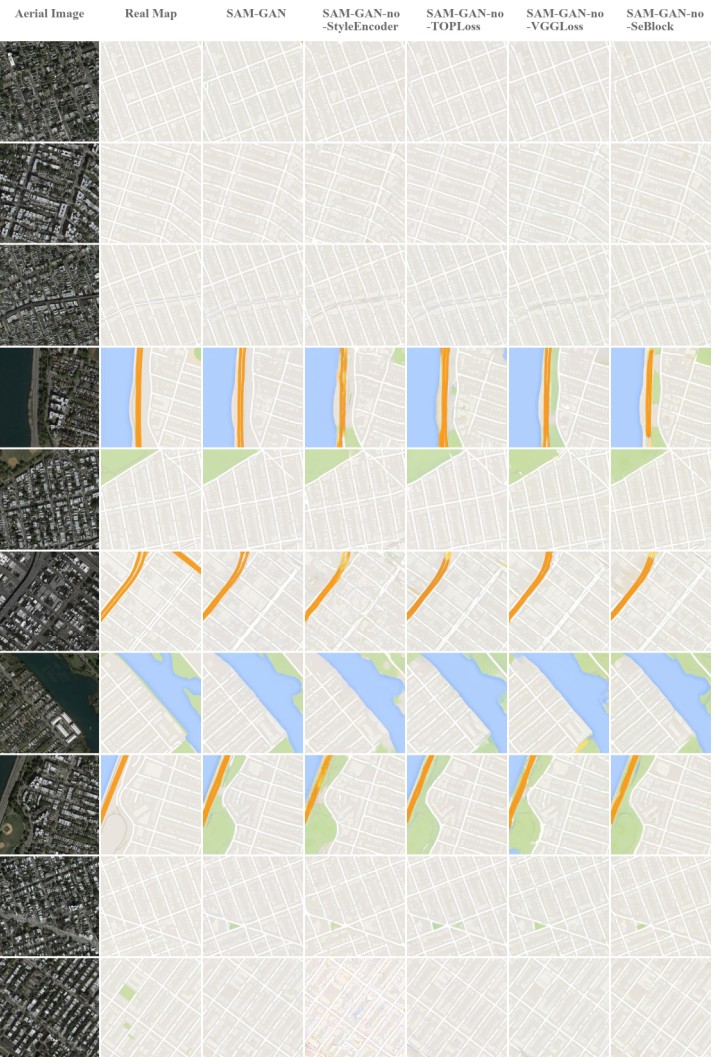

**Figure 6.** Qualitative comparison between aerial imagery and the map transformation model SAM-GAN in terms of ablation experiments. From left to right: aerial images, real maps, results of SAM-GAN, results of SAM-GAN-no-StyleEncoder, results of SAM-GAN-no-TOPLoss, results of SAM-GAN-no-VGGLoss, and results of SAM-GAN-no-SeBlock.

In summary, the primary role of the style encoder is to improve the image quality by learning the underlying coding space, while the VGG and topological losses are mainly used to improve the accuracy of feature boundary lines and the accuracy of the model for rendering feature colors in the generated maps, thus increasing the structural correlation between the generated maps and the real maps and also improving the visual performance of the generated results. The attention mechanism used in SAM-GAN enhances the learning ability of the model during the training process and accelerates the convergence of the model. Combining the qualitative and quantitative results shows that SAM-GAN generates high-quality maps while also enhancing the visual aspects of the images.

SAM-GAN provides better visual translation results from aerial images than SAM-GAN-no-StyleEncoder, SAM-GAN-no-VGGLoss, SAM-GAN-no-TOPLoss, and SAM-GAN-no-SeBlock, especially in terms of the edge detail of buildings in the image, completeness of roads, features, the rendering of corresponding colors, etc. This suggests that using the style encoder, the content loss, and the attention mechanism proposed in this paper can improve the model's ability to capture image details.

## 5. Discussion

SAM-GAN presents an end-to-end aerial image-to-map-based translation model, which performed well compared to some SOAT models. Although SAM-GAN can achieve better results than other image translation models in never-before-seen test sets, the results are not always positive. For example, when there are blurred areas in the aerial image itself and when there are colored areas in the image that makeup very little of the image as a whole, the model results may show inaccurate color rendering and blurred details. The above illustrates the model's shortcomings but simultaneously will prompt us to explore more options.

1.  The model's performance depends not only on the number of datasets but also on the data quality. We note that some of the aerial images in the training set appear to be in the same area but correspond to different features on the map. Moreover, the maps sometimes do not contain complete road information, so we speculate that improving the image quality of the dataset may improve the model's learning of feature classes and edge information.
2.  The SAM-GAN in this paper is a two-domain translation model. Given that the content and style encoders in the model generator do not interact with each other in learning information in the corresponding domains, the dataset can be expanded to use aerial images of different resolutions and map tiles of different styles in combination with each other to achieve multi-domain translation, which may enhance the generality of the model. However, as the number and variety of datasets increase, higher demands on the model's training time and hardware equipment are expected.
3.  More target domain styles can be used, not limited to map styles, e.g., art maps and game maps.

The current SAM-GAN model demonstrates its notable performance in generating maps using a novel mapping approach.

We also present a partial mechanism analysis of the methods and results of this paper from a cartographic perspective.

The traditional process of map mapping involves the acquisition of vectorized data and later processing by a large number of professional cartographers. This not only has strict requirements on the data source but also requires a lot of human resources and time, making the whole mapping process long and difficult to adapt to the needs of time-sensitive mapping in today's big data era. Since the 21st century, with the development and advancement of Internet information technology and visualization, the huge amount of data generated has become the driving force of the times, bringing new opportunities and challenges to various fields. The combination of digital information technology and cartography has led several innovative studies. Compared with the data used in traditional mapping, aerial remote sensing image data are richer in data type and volume and have significant advantages in terms of access, fineness, coverage, and real-time. Therefore, in this paper, we choose to use aerial remote-sensing images as the data source to improve the timeliness and convenience of map making.

As batch map generation is the current goal in aerial image and map translation, a deep learning model based on generative adversarial networks was chosen for this study, which is ideal for batch data generation. Based on the experimental results of this paper, the SAM-GAN method proposed in this paper can achieve the effect of processing a large amount of map data efficiently and accomplish the task of batch map generation excellently. Therefore, the research of the model based on aerial remote sensing image big data is beneficial to improve the efficiency of mapping and reduce the difficulty of mapping.

The model in this paper realizes the fitting of remote sensing images to map tile data, which can improve the clarity of map information expression and increase the information expression ability based on improving map mapping efficiency.

Firstly, compared to aerial remote sensing image data, map tiles can represent the information and categories of features more clearly. Users do not need professional knowl-

edge of image interpretation to recognize the information expressed in the map. The resulting maps greatly extend the range of map users with an unambiguous representation. Secondly, the spatial relationships between features in the map (orientation, inclusion, separation, and intersection, etc.), which can be expressed in the map utilizing different style symbols, colors, etc., make the spatial relationships between features clearer, the content more hierarchical, and the elements such as points, lines, and surfaces more readable. Finally, maps generated using aerial imagery can inherit the high timeliness. The high timeliness enables fast task execution and meets emergency response needs so that users can access the generated map products as soon as possible and understand the changes in spatial information, thus meeting personalized needs. Examples include law enforcement agencies' supervision of illegal structures and visualization tools for emergency response departments in the face of natural disasters. This will promote more personalized and intelligent map mapping applications in the future, prompting the development of the map mapping field in the direction of artificial intelligence.

### 6. Conclusions

This paper proposes SAM-GAN, a model for aerial image-to-map translation. We show that SAM-GAN produces better quality maps in terms of IS, FID, SSIM, and other evaluation metrics and that SAM-GAN, with the help of a style encoder, learns the style of the target domain well enough to guide the decoder to produce high-quality converted images. In addition, we use the attention mechanism, VGG loss, and topological consistency loss to improve the model's results at the visual level and speed up the model's convergence, helping it pay better attention to details.

In conclusion, SAM-GAN achieves better results in both objective and subjective evaluations compared to previous baseline models used for aerial image translation, which illustrates the effectiveness of this model in aerial image-to-map translation work, and also provides new thinking for future map mapping for areas such as personalized and intelligent map use.

We plan to expand the dataset in the next step by setting up more styles and resolutions to test the model. We will also improve the model's accuracy in the translation process and explore image translation in multiple domains.

**Author Contributions:** Conceptualization, methodology, formal analysis, investigation, and writing–original draft preparation, Jian Xu; supervision, project management, funding acquisition and resources, Hongwei Li; data curation, Xiaowen Zhou; validation and visualization, Chaolin Han and Bing Dong; writing—review and editing, Jian Xu and Hongwei Li. All authors have read and agreed to the published version of the manuscript.

**Funding:** This research is supported by the key project of National Natural Science Foundation of China—"Machine Map Theory and Modeling Method", whose support number is 42130112, and "Theory and Method of Map and Spatial Cognition under Human-Machine-Environment Collaboration" of High-level Talents Research Project of Zhengzhou University, whose support number is 135-32310276.

**Institutional Review Board Statement:** Not applicable.

**Informed Consent Statement:** Not applicable.

**Data Availability Statement:** The dataset used in this experiment is from the open-source dataset of the pix2pix model.

**Acknowledgments:** The author would like to thank the model and dataset contributed by the open source. At the same time, we thank the editors and reviewers for their valuable comments.

**Conflicts of Interest:** The authors declare no conflict of interest.

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
