# Peer review of "SAM-GAN: Supervised Learning-Based Aerial Image-to-Map Translation via Generative Adversarial Networks"

_ijgi, doi:10.3390/ijgi12040159_

Round 1

Reviewer 1 Report

Review of the manuscript (article) titled SAM-GAN: Supervised Learning-Based Aerial Image-to-Map Translation via Generative Adversarial Networks

Below are some suggestions for improving your article.

·        All references listed in References were cited in the text. However, some references are not listed according to the template from the Instructions for Authors.

o  In line 617 should be written Bengio, Y. instead of Bengio, Y.J.s.

o  In line 618 should be written In Proceedings of the 27th International Conference on Neural Information Processing Systems instead of Neural Information Processing Systems.

o  Some words (CANADA in line 620), surnames of authors (ZHANG and WU in line 620; CHEN, GUAN, CHEN in line 622;  LU in line 624; LI in line 627), journal titles (COMPUTER SYSTEMS APPLICATIONS in lines 620-621; ACTA AUTOMATICA SINICA in line 633) are written in capital letters.

o  Page range is missing in lines: 625, 633, 643, 647, 653, 655, 657, 670, 672, 681, 690, 695, 707, 714, 722. It would be good to check!

o  It is unnecessary - (doi:- 10.13203/j.whugis20190159 in line 626; - 43, - 321, doi:- 10.16383/j.aas.2017.y000003 in line 633).

o  Whether the papers 10, 11, 19, 24, 25, 26, 28, 36, 42 and 46 have been published in the proceedings?
It is necessary to state according to
Instructions for Authors In Proceedings of the Name of the Conference, Location of Conference, Country, Date of Conference (Day Month Year).

o In lines 644-645, 661, 664, 667 and 686 should be written In Proceedings of the … instead of In Proceedings of the Proceedings ...

o    Should it be written J.a.p.a. in lines 652, 654, 688, 690, 701, 707, 711?
It would be good to check!

o    In the text should be a blank space before the citation of all references, e.g. in line 38 should be written Liao Ke [1] instead of Liao Ke[1].

o  In line 64 should be written by Phillip Isola [9] instead of by Phillip Isol[9].

o  In line 71 should be written by Taeksoo Kim et al. [12] instead of by Taeksoo Kim et al.

o  In line 76 should be written by Yunjey Choi et al. [13] instead of by Yunjey Choi[13] et al.

o   In line 122 should be written Jun Gu et al. [30] instead of Gu Jun et al.[30]

·     In line 8 should be written [email protected] (H.L.); instead of [email protected];

·         In line 24 better to write image-to-map instead of  image-to-image.

·         In line 141 should be written In this section instead of In this chapter.

·        Figure 1 is unreadable, the letters are too small.

·     In line 142, the text (This is a figure. Schemes follow the same formatting.) should be deleted.

·        In Figure 3 and 6, it would be good to write Real map instead of Ground truth.

·        It would be good if there was a uniform writing of terms SAM-GAN-no-LVGG (it is written SAM-GAN-no-VGGLoss) and SAM-GAN-no-LTop (it is written SAM-GAN-no-TOPLoss) in Figure 6 as in Tables 6 and 7 and in the text.

·     In line 556 should be written results of SAM-GAN-no-LVGG, results of SAM-GAN-no-LTop instead of results of SAM-GAN-no-LVGG, results of SAM-GAN-no-LTop.

·         Why is it not mentioned results of SAM-GAN below the Figure 6?

·     Should the order be swapped in Figure 6 SAM-GAN-no-TOPLoss (should be written SAM-GAN-no-LTop) and SAM-GAN-no-VGGLoss (should be written SAM-GAN-no-LVGG)?
It would be good to check!

·        Change the order of sections 5 and 6, i.e. instead of 5. Conclusions and 6. Discussion it should be 5. Discussion and 6. Conclusions.

Reviewer 2 Report

This paper proposes a supervised model (SAM-GAN) based on generative adversarial networks (GAN) to improve the performance of aerial image-to-map translation. Furthermore, using the Maps dataset, a comprehensive qualitative and quantitative comparison is made between the SAM-GAN model and previous methods used for aerial image-to-map translation in combination with excellent evaluation metrics.

Below are some of my specific comments.

(1) Lines 46-48: This is too simplistic to capture the specific problems with the current research. What are the current problems that are not solved or where is the room for improvement in current research?

(2) Lines 132-139: I didn't understand this part, which felt irrelevant to the article.

(3) Figure 1(a): The aerial image is too fuzzy. It is suggested to enlarge the ground objects to reduce the number of ground objects in the aerial image or improve the image quality to make the aerial image look clearer.

(4) Figure 1(b): The quality of the picture is too low and the marks in it are ambiguous.

(5) It is suggested to combine Figure(a) and (b) to form a picture, and add some words in the picture to help reading and understanding. Please do not only mark some custom symbols on the picture.

(6) Figure 4: There is no value when K=10 in Figure 4, which is inconsistent with the text.

(7) Line 481, "targets" is not clear, If "the number of targets" means K, then you should write it directly.

(8) The authors mention in line 495 that "the model performs best with a K range of 1-5", but the authors only study five cases where K is 1,3, 5, 8 and 10. What's more, Figure4 and Figure5 do not show any obvious difference in their quantitative evaluation results when K is different. If considering the comprehensive time, why not directly determine that K=1? Why do the authors write “1-5”?

(9) In the paper, the authors did not mention the relevant information of the dataset, such as resolution or image content, etc. I only saw relatively regular features such as water bodies, roads and buildings in the pictures. Is the model applicable to some complex features? What about the recognition effect?

(10) I suggest that Discussion be put at the front of the conclusion.

Round 2

Reviewer 3 Report

I agree with the author's response and modifications. But I still have some questions about experimental data:

1) Was the laboratory dataset constructed by the author of this article or by other scholars?

2) In this response, it was mentioned that other scholars have also used this dataset for research. My understanding is that this dataset should be publicly available, so the dataset needs to have reference citations. The literature review needs to introduce the work of relevant scholars. If it is related to this study, comparative analysis is also indispensable.

Author Response

Dear Reviewers, please see the attachment. Please allow us to express our gratitude to you. Thank you for your constructive comments on our paper, which were very important to us.
